# Effects of Bafa Wubu and He-Style Tai Chi exercise training on physical fitness of overweight male university students: A randomized controlled trial

Yantao Niu[1,2☯], Rojapon Buranarugsa[1,3☯]*, Piyathida Kuhirunyaratn[4]

**1** Exercise and Sport Sciences Program, Graduate School, Khon Kaen University, Khon Kaen, Thailand, **2** Faculty of Physical Education, Jiaozuo Normal College, Jiaozuo, China, **3** Physical Education Program, Faculty of Education, Khon Kaen University, Khon Kaen, Thailand, **4** Department of Community Medicine, Faculty of Medicine, Khon Kaen University, Khon Kaen, Thailand

☯ These authors contributed equally to this work.
* rojapon@hotmail.com

**Data Availability Statement:** All relevant data are within the manuscript and its Supporting Information files.

## Abstract

This study aimed to compare the effects of 12-week Bafa Wubu Tai Chi (BW-TC) and traditional He-style Tai Chi (TH-TC) exercise training on physical fitness parameters in overweight university students and to compare the differences in their effects. A total of 81 overweight male university students were randomly assigned to the BW-TC group (N = 27), the TH-TC group (N = 27), and the control group (CG, N = 27). Upper limb grip strength, wall squat, sitting and reaching, 6-minute walk, single-leg stance, and Y-balance were measured at baseline and after 12 weeks of Tai Chi training. There were no significant differences in demographic characteristics and assessment parameters among the groups at baseline ($p$>0.05). Both BW-TC and TH-TC performed Tai Chi exercise program training lasting 12 weeks, with three sessions per week, each lasting 60 minutes intervention. The changes in mean scores for the sit-and-reach test were 3.11 cm and 4.52 cm, for the wall squat test were 27.56 s and 36.85 s, and for the 6-minute walk test were 22.93 m and 63.22 m, and Y-balance ($p$<0.05) significantly increased in both BW-TC and TH-TC groups, while the mean score of single-leg stance significantly decreased ($p$<0.05). Additionally, compared to the BW-TC group, the TH-TC group showed a significant increase in lower limb strength (13.89 s, $p$ = 0.048) and the distance of the Y-balance test in the left posterior medial direction (4.04 cm, $p$ = 0.031). BW-TC and TH-TC interventions effectively improved physical fitness in overweight university students. However, TH-TC showed superior results in lower limb strength improvement.

**Trial registration number:** ChiCTR2200059427 (https://www.chictr.org.cn).

## Introduction

Tai Chi is a popular form of exercise that has been practiced in China for centuries, which has gained popularity worldwide in recent years due to its numerous health benefits. This ancient form of exercise is known to improve physical and mental well-being by incorporating

**Funding:** This research was supported by the 2022 Key University Research Project of the Department of Education of Henan Province, China (Project Number: 22B890001.

**Competing interests:** The authors have declared that no competing interests exist.

mindfulness, breathing techniques and gentle movements [1, 2]. The characteristics of Tai Chi exercises can benefit the health of individuals who are overweight or obese, which can improve posture and hand flexibility and enhance lower limb muscle strength, coordination and dynamic balance [3, 4]. Tai Chi exercise has been found to have significant effects on improving the physical fitness of overweight or obese individuals. Participants practice Tai Chi in an upright posture with continuous slow movements, shifting their body weight from side to side while maintaining their balance [5]. Previous studies conducted by international and domestic scholars have reported that Tai Chi exercise can lead to improvements in body composition [6, 7]; lower-limb strength [4, 8–11]; endurance [12]; balance [3, 11, 13–18]; and other aspects of physical health in overweight and obese individuals.

Despite the many benefits of Tai Chi, there is limited research on the health benefits of different styles thereof—specifically Bafa Wubu Tai Chi (BW-TC) and traditional He-style Tai Chi (TH-TC)—for overweight practitioners. While previous research on this form of exercise has mostly focused on the intervention benefits of simplified or modified forms of Tai Chi—such as 24-form Tai Chi [19]; 42-form Tai Chi [20]; Chen-style Tai Chi [21]; and modified Tai Chi—limited research has been conducted on the fitness benefits of traditional Tai Chi [22]. Because the health enhancement and appeal of traditional Tai Chi continue to attract different age groups, the advantages of practicing traditional Tai Chi to improve overall fitness should be further explored.

BW-TC is a simplified Tai Chi routine that has been recognised and promoted by the General Administration of Sports in China. Its movement characteristics include simple movements, suitable postures and ease-of-learning. The routine consists of a 16-movement set that takes approximately three minutes to complete [23]. There has been an increasing amount of research on the benefits of BW-TC, which can not only help to alleviate anxiety and depression in college students through exercise [24, 25]; but also improves lower limb control and strengthens support [10].

In contrast, TH-TC represents the traditional form of Tai Chi combined with the eight hand and five footwork methods of BW-TC. TH-TC movements are characterised by a combination of these techniques; body movements are based on a standing circle, with the body's centre-of-gravity continuously shifting up and down. It takes approximately 5–8 minutes to complete the 72-movement routine [26]. While no previous studies have reported benefits of TH-TC, to our knowledge, no studies have investigated the effects of BW-TC and TH-TC on obese and overweight individuals. As such, the aims of this study are to investigate the effects of two styles of Tai Chi training on the physical fitness of overweight college students and to compare the differences in these effects.

## Materials and methods

### Ethical approval

This study was approved by the Khon Kaen University Ethical Committee for Human Research (approval no. HE642132; see S1 File); and the trial was registered with the Chinese Clinical Trial Registry (registration no. ChiCTR2200059427) and carried out in accordance with the Declaration of Helsinki. All participants signed a consent form and were informed of the purpose of the study in writing, the procedures involved and the benefits of participating in this exercise program.

### Participants

This study was to evaluate the effect of two different types of tai chi on the physical fitness of overweight male college students. The previous research literature has no similar groups,

exercise intensity and parameters. Therefore, G*Power (Version 3.1.9.4) was used to estimate the required sample size. The F test with 'ANOVA: Fixed effects, omnibus, one-way' was selected, and 'a priori: Compute required sample size' was used for the analysis. The effect size convention was set to the maximum value of f = 0.40, the significance level ($\alpha$ err prob) was set to 0.05, the desired power value (1 - $\beta$ err prob) was set to 0.80, and the number of groups was set to 3. The initial estimated sample size was 66 participants, with 22 participants in each group. To account for a potential 20% dropout rate, 27 participants per group were required, resulting in a total of 81 participants; we increased the sample size to 30 participants per group to enhance the statistical power of the study, yielding a total of 90 participants.

Due to the COVID-19 pandemic, only male college students from Jiaozuo Normal University in Jiaozuo City, Henan Province, China were recruited for this study from September 26, 2021. Recruitment was carried out via posters placed in the college student activity centre and sports field. A total of 123 male college students between 18–23 years of age participated in the qualification screening, and 90 eligible volunteers were selected to participate in the study; nine volunteers withdrew during the study, however, resulting in a final sample size of 81 participants who completed the entire research process. Inclusion criteria included male college students 18–23 years of age in their first or second year at university with BMIs between 24.0–27.9 (i.e., the Chinese overweight standard), who provided informed consent and voluntarily participated. Long-term Tai Chi athletes, sports association members, individuals with severe cardiovascular or musculoskeletal system diseases and those with a BMI below 24 or greater than 27.9 were excluded. Fig 1 presents a flow chart outlining the participants' entire intervention process and the effects of progress in this trial, which was designed, analysed and interpreted according to the Consolidated Standards for Reporting Trials criteria (see S2 and S3 Files).

## Experimental procedure

The project manager for this randomised controlled-trial experimental study utilised web programming on the Researcher Randomizer website (http://www.randomizer.org) to randomly divide the participants into the BW-TC group, the TH-TC group and the control group (CG). The BW-TC and TH-TC groups received Tai Chi training three times a week for 12 weeks; each class lasted 60 minutes and included 15 minutes of warm-up exercises, 30 minutes of Tai Chi and 15 minutes of cool-down exercises. The control group received health lectures once a month for three months, but did not exercise.

The intervention period was from October 18, 2021 to January 7, 2022. During the experiment, all three groups maintained their normal diet and daily lives and adhered to COVID-19 requirements by wearing masks, sanitising and maintaining social distancing. Training was scheduled during the students' daily activity time on Mondays, Wednesdays and Fridays from 5:30 pm to 6:30 pm. All assessments were conducted on Saturdays and Sundays, and the exercise programs were performed under the direction of researchers and trained research assistants. Before the intervention, all participants underwent a baseline test, which included an evaluation of basic personal physical signs such as their gender, ages, heights, weights and BMI values; the participants' flexibility, upper- and lower-limb strength, balance and aerobic endurance were also evaluated. After 12 weeks of Tai Chi training, the remaining 81 participants were re-assessed with the same variables as before the intervention. The protocol of this trial is available as S2 File.

## Intervention

**BW-TC group.** The BW-TC group was used for intervention. The movements of this form are known for their simplicity, reasonable posture and ease-of-learning and -practice,

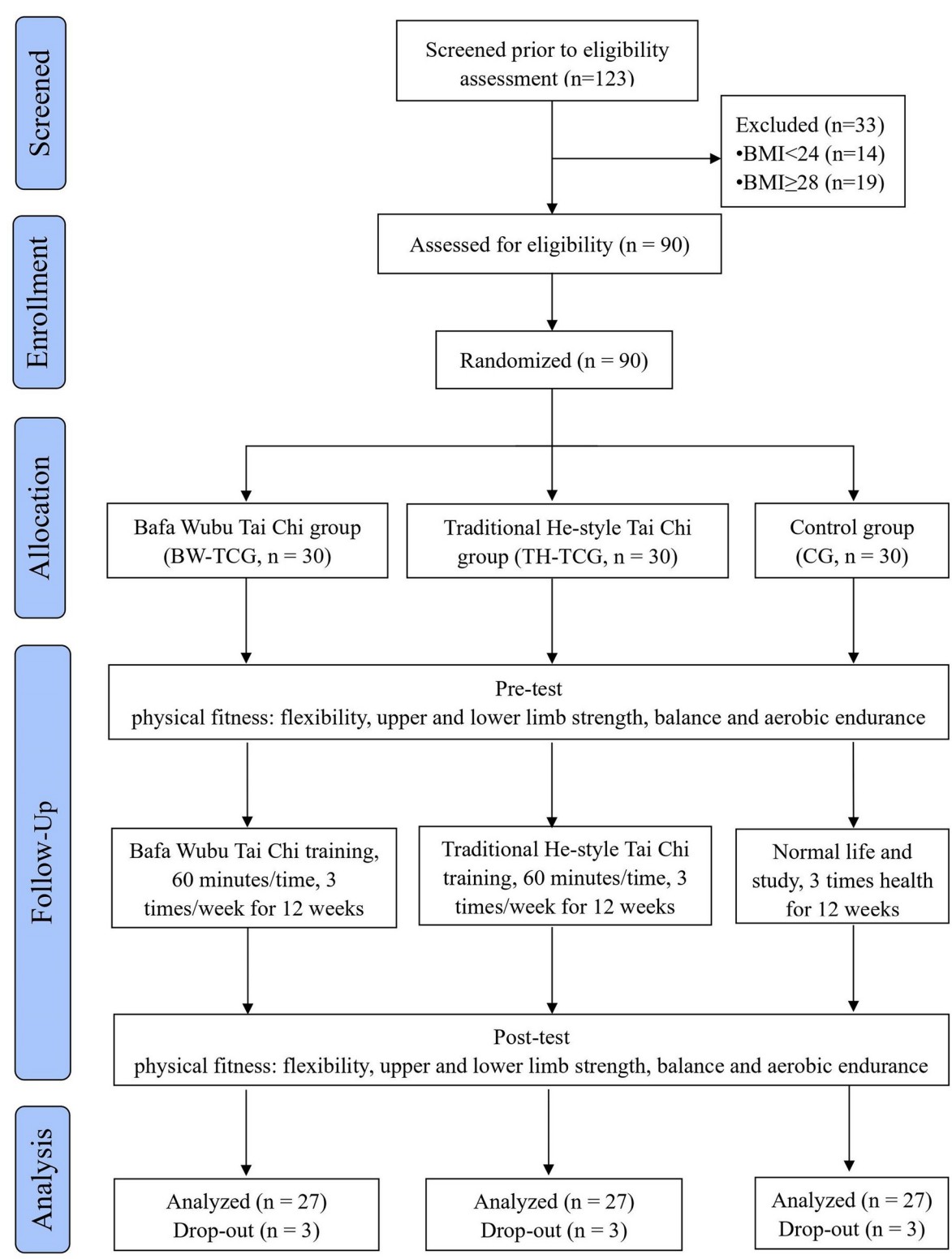

**Fig 1. CONSORT flow diagram.**

with each set taking approximately three minutes to complete [23]. The participants were led through a 60-minute class by a qualified Tai Chi instructor, who provided instructional guidance for 30 minutes of continuous BW-TC practice and 15 minutes of warm-up and relaxation exercises and emphasised the importance of caution during the practice.

**TH-TC group.** The TH-TC group was also used for intervention, and The Book of He Style Taijiquan by Youlu He (2003) was selected for the instructional content. TH-TC includes 72 circle and arc movements, which cause the effect of gravity on the body to continuously fluctuate [27]; it takes 5–8 minutes to complete one set. A qualified Tai Chi instructor provided instructional guidance and emphasised the importance of caution throughout the 60-minute class, which included 15 minutes of warm-up exercises, 30 minutes of continuous TH-TC practice and 15 minutes of relaxation exercises.

**Control group.** While the control group did not receive any exercise intervention, the participants were asked to attend three health lectures held on Monday afternoons during the first, fifth and ninth weeks after the start of the experiment, while maintaining their normal daily routines.

## Outcome measures

**Flexibility test.** Flexibility was assessed using the sit-and-reach test using the WQ-168 sitting-body forward-bending tester (China's Beijing Weixin Yiao Technology Development Company, n.d.). Participants sat on a mat with their legs stretched out straight and feet together and were instructed to bend forward and push a test cursor with their fingertips [28]; the distances they reached was measured in centimetres, and the test was performed twice to record the best result.

**Upper limb strength test.** Using an HKD-1881 electronic grip strength tester (manufactured by Beijing Hongkangda Sports Technology Co., Ltd.), participants stood in a natural position with their feet hip-width apart and gripped the device in their dominant hand while their arms hung naturally at their sides [29]. This test was performed twice, and the participants' grip strength was measured in kilograms to one decimal point; the highest score was recorded for the data analysis.

**Lower limb strength test.** During the wall-squat test, the participants stood with their feet shoulder-width apart and leaned back against a vertical wall, then pressed their backs against the wall as they slid down to a seated position with their knees even with their hips, their thighs parallel to the floor and their arms hanging naturally at their sides [30]. The total time each participant could hold this isometric position was recorded, and the test was repeated twice, with the best score used for data analysis.

**Balance.** For static balance assessment, the Balance Error Scoring System (BESS) was used, which Riemann et al. (1999) observed had high reliability for measuring static balance, with a 0.78–0.96 within-class correlation coefficient [31]. The procedure consists of three balancing postures measured on stable ground and on a soft cushion; the participants were asked to maintain each posture with their eyes closed and their hands on their hips for 20 seconds, and the number of loss of balance recorded [32]. The Y-Balance Test (YBT)—which Col et al. (2013) found had a high reliability of 0.976–0.982—was used to provide a dynamic balance assessment [33]. The participants were asked to balance on one foot on a test platform and use their other foot to push a test board in different directions as far as possible without support, and their furthest distances were recorded; according to Johnston et al. (2018), this method is accurate to 0.5cm [34]. Each test was repeated twice, with the best score recorded for data analysis.

**Six Minute Walk Test.** The Six-Minute Walk Test (6MWT) was used to measure aerobic endurance. Participants walked along a 30-metre-long line with cones at both ends and were

instructed to cover the longest distance possible within six minutes, chairs were provided along the trail for resting, and the distance each participant walked in six minutes was recorded in metres [35].

**Statistical analysis.**   Data were collected and compiled on an Excel 2019 spreadsheet. Statistical analyses were carried out using IBM® SPSS® Statistics 26.0 software (IBM Corporation, NY, USA). Data were presented as mean ± standard deviation (SD), and a 95% confidence interval (CI) was adopted. The Shapiro¬–Wilk test checks for consistency and normal distribution of data. Descriptive statistics were applied to summarize the baseline characteristics of the participants, and one-way ANOVA tests were used to compare differences in the groups' baselines. Following the 12th-week data collection, we conducted a mixed design analysis of variance to explore the effect of intervention and control groups on the dependent variables (flexibility, upper and lower limb strength, balance assessments, and the 6-minute walk test). This analysis focused on exploring the effects of time (pre- and post-intervention) and group (different intervention and control groups) on the outcomes. Upon identifying significant interaction effects, we further used simple time effects and simple time by group effects to comprehensively expound upon the differences among various groups before and after the intervention and following the 12-week intervention. The significance level was set at $p < 0.05$.

## Results

In this study, nine participants voluntarily withdrew due to the impact of the COVD-19, finally the influencing factors and trends of evaluated variables were analysed in a total of 81 overweight male college students aged 18–23 years old with 24.0–27.9 BMIs, across three groups and two time points (see S4 File). There were no statistically significant differences between the demographic variables (i.e., mean age, weight, height and BMI) of the three groups, and no statistically significant differences were detected between the groups' baselines (see Table 1).

Our study's statistical analysis unveiled significant interactions between time effects and time-by-group effects across multiple variables, encompassing flexibility, lower limb strength, Single-leg stand (on Stable ground and Soft cushion), Tandem stand (on Stable ground and Soft cushion), Y-balance test (for both Right-leg and Left-leg: Anterior, Posteromedial, and Posterolateral), and the six-minute walk test ($p < 0.05$). It is worth noting that no significant interaction was detected in the context of upper-limb strength ($p > 0.05$; see Table 2). The researchers subsequently conducted a detailed analysis of the simple time effect and the simple time-by-group effect to compare the differences of each group before and after the intervention, as well as differences between the three groups following the 12-week intervention.

After 12 weeks of intervention, all assessment items—flexibility, lower-limb strength and the Y-balance test—and the 6WMT increased significantly ($p < 0.005$) for the BW-TC and TH-TC groups; while all BESS assessment items significantly decreased ($p < 0.001$). No statistically significant differences were observed in the CG's physical fitness assessment items (see Table 3).

Compared to the CG, the BW-TC group significantly improved their flexibility ($p < 0.001$); lower-limb strength ($p < 0.001$); anterior right leg during YBT ($p = 0.002$); posteromedial right leg during YBT ($p = 0.001$); posterolateral right leg during YBT ($p < 0.001$); anterior left leg during YBT ($p = 0.002$); posterolateral left leg during YBT ($p < 0.001$); and 6MWT ($p = 0.009$). Notably, however, this group demonstrated significant decreases in single-leg stands on a soft cushion ($p = 0.005$); tandem stands on stable ground ($p = 0.022$); and tandem stands in a soft cushion ($p = 0.008$). The TH-TC group also significantly increased flexibility ($p < 0.001$); lower limb strength ($p < 0.001$); all YBT assessment items ($p < 0.001$); and

**Table 1. Demographic, baseline body composition and physical fitness of study participants.**

| Parameters | BWTCG (n = 27) (mean ± SD) | THTCG (n = 27) (mean ± SD) | CG (n = 27) (mean ± SD) | p-value |
|---|---|---|---|---|
| Age (year) | 18.48 ± 0.64 | 18.41 ± 0.75 | 18.67 ± 0.62 | 0.350 |
| Height (cm) | 175.56 ± 5.19 | 174.41 ± 5.28 | 175.41 ± 4.14 | 0.646 |
| Bodyweight (kg) | 80.33 ± 5.81 | 80.50 ± 5.33 | 80.66 ± 4.29 | 0.974 |
| BMI (kg/m$^2$) | 26.07 ± 1.30 | 26.45 ± 1.12 | 26.23 ± 1.12 | 0.496 |
| Flexibility (cm) | 12.15±4.06 | 11.63±3.59 | 11.30±4.99 | 0.760 |
| Upper limb strength (kg) | 41.22±4.53 | 41.81±6.31 | 45.04±7.48 | 0.059 |
| Lower limb strength (second) | 106.26 ± 22.66 | 110.85 ± 28.25 | 105.67 ± 11.87 | 0.640 |
| Single-leg stand-Stable ground (0–10 score) | 2.37 ± 1.92 | 1.85 ± 0.77 | 1.63 ± 1.18 | 0.135 |
| Single-leg stand-Soft cushion (0–10 score) | 3.48 ± 1.60 | 2.67 ± 1.27 | 3.15 ± 1.32 | 0.108 |
| Tandem stand-Stable ground (0–10 score) | 1.85 ± 1.17 | 1.48 ± 0.70 | 1.56 ± 0.75 | 0.282 |
| Tandem stand-Soft cushion (0–10 score) | 2.48 ± 0.85 | 2.44 ± 0.75 | 2.11 ± 0.97 | 0.228 |
| YBT(Right-leg)-Anterior (cm) | 61.93 ± 6.12 | 60.81 ± 5.05 | 61.19 ± 6.79 | 0.789 |
| YBT(Right-leg)-Posteromedial (cm) | 101.07 ± 7.68 | 104.33 ± 6.09 | 101.44 ± 7.07 | 0.178 |
| YBT(Right-leg)-Posterolateral (cm) | 93.26 ± 6.56 | 94.52 ± 6.91 | 93.04 ± 7.53 | 0.705 |
| YBT(Left-leg)-Anterior (cm) | 62.22 ± 4.86 | 61.74 ± 4.71 | 62.70 ± 6.84 | 0.817 |
| YBT(Left-leg)-Posteromedial (cm) | 100.81 ± 8.85 | 102.04 ± 6.75 | 101.15 ± 5.72 | 0.814 |
| YBT(Left-leg)-Posterolateral (cm) | 93.00 ± 8.20 | 93.22 ± 6.86 | 94.44 ± 5.11 | 0.707 |
| 6MWT (meter) | 414.59 ± 41.41 | 389.67 ± 70.08 | 395.85 ± 55.58 | 0.252 |

Note: One-way ANOVA test

BWTCG, Bafa Wubu Tai Chi group; THTCG, Traditional He-style Tai Chi group; CG, Control group; BMI, body mass index; YBT, Y-balance test; 6MWT, six-minute walk test; cm, centimeter; kg, Kilogram; m, meter.

6MWT ($p$ = 0.001). Compared to the CG, this group also demonstrated significantly decreased single-leg stands on stable ground ($p$ = 0.028); single-leg stands on a soft cushion ($p$ < 0.001); tandem stands on stable ground ($p$ = 0.002); and tandem stands on a soft cushion ($p$ < 0.001). Other than significant increases observed in lower-limb strength ($p$ = 0.048) and posteromedial left leg during YBT ($p$ = 0.031), when comparing the TH-TC and BW-TC groups, no statistically significant differences were observed in any of the other assess items (see Table 4).

Furthermore, based on our previous research, we observed that these overweight male college students after the 12-week intervention, the BW-TC group and TH-TC group showed a significant decrease in body weight by 2.69 kg and 2.04 kg, respectively, and in BMI by 0.90 kg/m$^2$ and 0.67 kg/m$^2$, respectively, compared to baseline measurements at week zero. When compared to the CG, the BW-TC group exhibited significant decreases in body weight ($p$ = 0.031) and BMI ($p$ = 0.001), while no significant differences were observed in the TH-TC group. Additionally, there were no significant differences in body weight and BMI improvements between the two Tai Chi groups [36].

## Discussion

After 12 weeks of Tai Chi training, both intervention groups showed significant improvements in flexibility, lower-limb strength and the 6WMT, static-balance and dynamic-balance tests, compared to the control group, while there were no significant changes in the mean scores of the study participants' upper-limb strength; interestingly, the TH-TC group performed better than the BW-TC group in the lower-limb strength test and the posteromedial left-leg balance ability evaluation in the YBT test. This echoes several studies, which concluded that Tai Chi improves balance, muscle mass, endurance, flexibility, muscle strength and balance control

**Table 2. Baseline and after intervention mean and standard deviation, and *f* statistic for time effects and time by group effects, *p*-values, and partial eta squared of parameters of three groups.**

| Parameters | BWTCG (n = 27) (Mean ± SD) | | THTCG (n = 27) (Mean ± SD) | | CG (n = 27) (Mean ± SD) | | Time effects | *p*-value | PES | Time*Group effects | *p*-value | PES |
|---|---|---|---|---|---|---|---|---|---|---|---|---|
| | Pre | Post | Pre | Post | Pre | Post | | | | | | |
| Flexibility (cm) | 12.15 ± 4.06 | 15.26 ± 3.61 | 11.63 ± 3.59 | 16.15 ± 2.82 | 11.30 ± 4.99 | 11.48 ± 4.69 | $F_{(1,78)} = 262.244$ | < .001 | 0.771 | $F_{(2,78)} = 62.950$ | < .001 | 0.617 |
| ULS (kg) | 41.22 ± 4.53 | 41.37 ± 4.23 | 41.81 ± 6.31 | 41.37 ± 5.66 | 45.04 ± 7.48 | 44.59 ± 7.19 | $F_{(1,78)} = 1.455$ | 0.231 | 0.018 | $F_{(2,78)} = 0.931$ | 0.398 | 0.023 |
| LLS (second) | 106.26 ±22.66 | 133.81 ±24.84 | 110.85 ±28.25 | 147.70 ±34.28 | 105.67 ±11.87 | 107.00 ±11.82 | $F_{(1,78)} = 211.955$ | < .001 | 0.731 | $F_{(2,78)} = .49.915$ | < .001 | 0.561 |
| SLS-SG (0–10 score) | 2.37 ± 1.93 | 1.48 ± 1.48 | 1.85 ± 0.77 | 1.00 ± 0.62 | 1.63 ± 1.18 | 1.70 ± 1.20 | $F_{(1,78)} = 23.235$ | < .001 | 0.230 | $F_{(2,78)} = 7.470$ | .001 | 0.161 |
| SLS-SC (0–10 score) | 3.48 ± 1.60 | 2.00 ± 1.71 | 2.67 ± 1.27 | 1.30 ± 0.78 | 3.15 ± 1.32 | 3.04 ± 1.34 | $F_{(1,78)} = 56.178$ | < .001 | 0.419 | $F_{(2,78)} = 11.122$ | < .001 | 0.222 |
| TS-SG (0–10 score) | 1.85 ± 1.17 | 1.00 ± 0.78 | 1.48 ± 0.70 | 0.81 ± 0.62 | 1.56 ± 0.75 | 1.48 ± 0.85 | $F_{(1,78)} = 36.146$ | < .001 | 0.317 | $F_{(2,78)} = 7.057$ | 0.002 | 0.153 |
| TS-SC (0–10 score) | 2.48 ± 0.85 | 1.52 ± 0.85 | 2.44 ± 0.75 | 1.22 ± 0.58 | 2.11 ± 0.97 | 2.19 ± 1.18 | $F_{(1,78)} = 68.790$ | < .001 | 0.469 | $F_{(2,78)} = 21.787$ | < .001 | 0.358 |
| YBT(RL)-A (cm) | 61.93 ±6.12 | 66.19 ±5.34 | 60.81 ±5.05 | 68.74±5.46 | 61.19 ±6.79 | 61.04 ±6.46 | $F_{(1,78)} = 130.327$ | < .001 | 0.626 | $F_{(2,78)} = 44.102$ | < .001 | 0.531 |
| YBT(RL)-PM (cm) | 101.07±7.68 | 108.70±7.09 | 104.33±6.09 | 111.67±6.62 | 101.44±7.07 | 102.15±6.95 | $F_{(1,78)} = 109.334$ | < .001 | 0.584 | $F_{(2,78)} = 20.493$ | < .001 | 0.344 |
| YBT(RL)-PL (cm) | 93.26±6.56 | 102.93±7.46 | 94.52±6.91 | 103.04±8.33 | 93.04±7.53 | 94.48 ± 6.66 | $F_{(1,78)} = 122.540$ | < .001 | 0.611 | $F_{(2,78)} = 18.917$ | < .001 | 0.327 |
| YBT(LL)-A (cm) | 62.22 ± 4.86 | 67.59 ± 4.41 | 61.74 ± 4.71 | 69.52 ± 5.12 | 62.70 ± 6.84 | 62.81 ± 6.39 | $F_{(1,78)} = 109.255$ | < .001 | 0.583 | $F_{(2,78)} = 28.659$ | < .001 | 0.424 |
| YBT(LL)-PM (cm) | 100.81±8.85 | 105.67±7.57 | 102.04±6.75 | 109.70±6.88 | 101.15±5.72 | 102.67±5.64 | $F_{(1,78)} = 63.388$ | < .001 | 0.448 | $F_{(2,78)} = 9.142$ | < .001 | 0.190 |
| YBT(LL)-PL (cm) | 93.00±8.20 | 102.44±7.61 | 93.22±6.83 | 104.81±5.43 | 94.44±5.11 | 94.48±4.02 | $F_{(1,78)} = 202.428$ | < .001 | 0.722 | $F_{(2,78)} = 51.652$ | < .001 | 0.570 |
| 6MWT (meter) | 414.59 ±41.41 | 437.52 ±49.09 | 389.67 ±70.08 | 452.89 ±74.90 | 395.85 ±55.58 | 393.48 ±53.94 | $F_{(1,78)} = 32.852$ | < .001 | 0.296 | $F_{(2,78)} = 15.367$ | < .001 | 0.283 |

Note: Mixed design ANOVA test; Adjustment for multiple comparisons: Least Significant Difference.

BWTCG, Bafa Wubu Tai Chi group; THTCG, Traditional He-style Tai Chi group; CG, Control group; PES, Partial Eta Squared; ULS, Upper limb strength; LLS, Lower limb strength; SLS-SG, Single-leg stand-Stable ground; SLS-SC, Single-leg stand-Soft cushion; TS-SG, Tandem stand-Stable ground; TS-SC, Tandem stand-Soft cushion; YBT, Y-balance test; YBT(RL)-A, YBT(Right-leg)-Anterior; YBT(RL)-PM, YBT(Right-leg)-Posteromedial; YBT(RL)-PL, YBT(Right-leg)-Posterolateral; YBT(LL)-A, YBT(Left-leg)-Anterior; YBT(LL)-PM, YBT(Left-leg)-Posteromedial; YBT(LL)-PL, YBT(Left-leg)-Posterolateral; 6MWT, six-minute walk test. cm, centimeter; kg, Kilogram.

[16, 37–40]. Our study involved 12 weeks of 60-minute Tai Chi training sessions conducted three times every week for overweight male college students and achieved similar results.

The current study found that the mean differences in flexibility before and after the BW-TC and TH-TC groups were −3.1 cm and −4.52 cm, representing an increase of 25.6% and 38.8%, respectively. Tai Chi requires practitioners' spine to be straight and involves switching between bow stance and horse stances, which helps exercise the stretchability of trunk ligaments and the flexibility of hamstrings. Our finding was in line previous research, which found that Tai Chi significantly improved flexibility in older Chinese individuals with risk factors for cardiovascular disease [41]. Notably, however, multiple comparisons revealed no significant difference between the flexibility mean scores of the BW-TC and TH-TC groups after 12 weeks. Even though the two styles of Tai Chi evaluated in this study are different, further research with a longer-duration programme is needed to explore the differences in the effects thereof on flexibility.

**Table 3. Mean differences, *p*-value, and 95% confidence interval for differences in paired t-test before and after 12 weeks of intervention within each group.**

| Parameters | BWTCG Md (95% CI) | *p*-value | THTCG Md (95% CI) | *p*-value | CG Md (95% CI) | *p*-value |
|---|---|---|---|---|---|---|
| Flexibility (cm) | -3.11(-3.67 to -2.56) | < .001 | -4.52(-5.07 to -3.96) | < .001 | -0.19(-0.74 to 0.37) | 0.508 |
| Upper limb strength (kg) | -0.15(-0.84 to 0.56) | 0.677 | 0.44(-0.26 to 1.15) | 0.214 | 0.44(-0.26 to 1.15) | 0.214 |
| Lower limb strength (second) | -27.56(-32.75 to -22.37) | < .001 | -36.85(-42.04 to -31.66) | < .001 | -1.33(-6.52 to 3.86) | 0.610 |
| Single-leg stand-Stable ground (0–10 score) | 0.89(0.49 to 1.29) | < .001 | 0.85(0.45 to 1.25) | < .001 | -0.07(-0.47 to 0.32) | 0.712 |
| Single-leg stand-Soft cushion (0–10 score) | 1.48(1.03 to 1.94) | < .001 | 1.37(0.92 to 1.83) | < .001 | 0.11(-0.34 to 0.57) | 0.628 |
| Tandem stand-Stable ground (0–10 score) | 0.85(0.55 to 1.16) | < .001 | 0.67(0.36 to 0.97) | < .001 | 0.07(-0.23 to 0.38) | 0.630 |
| Tandem stand-Soft cushion (0–10 score) | 0.96(0.67 to 1.26) | < .001 | 1.22(0.93 to 1.52) | < .001 | -0.07(-0.37 to 0.22) | 0.616 |
| YBT(Right-leg)-Anterior (cm) | -4.26(-5.47 to -3.05) | < .001 | -7.93(-9.14 to -6.71) | < .001 | 0.15(-1.06 to 1.36) | 0.808 |
| YBT(Right-leg)-Posteromedial (cm) | -7.63(-9.35 to -5.91) | < .001 | -7.33(-9.06 to -5.61) | < .001 | -0.70(-2.43 to 1.02) | 0.418 |
| YBT(Right-leg)-Posterolateral (cm) | -9.67(-11.71 to -7.63) | < .001 | -8.52(-10.56 to -6.4) | < .001 | -1.44(-3.48 to 0.59) | 0.162 |
| YBT(Left-leg)-Anterior (cm) | -5.37(-6.83 to -3.91) | < .001 | -7.78(-9.24 to -6.32) | < .001 | -0.11(-1.57 to 1.35) | 0.880 |
| YBT(Left-leg)-Posteromedial (cm) | -4.85(-6.88 to -2.83) | < .001 | -7.67(-9.69 to -5.64) | < .001 | -1.52(-3.55 to 0.51) | 0.140 |
| YBT(Left-leg)-Posterolateral (cm) | -9.44(-11.15 to -7.74) | < .001 | -11.59(-13.30 to -9.89) | < .001 | -0.04(-1.74 to 1.67) | 0.966 |
| 6MWT (meter) | -22.93(-39.73 to -6.13) | 0.008 | -63.22(-80.02 to -46.42) | < .001 | 2.37(-14.43 to 19.17) | 0.780 |

Note: paired t-test.

BWTCG, Bafa Wubu Tai Chi group; THTCG, Traditional He-style Tai Chi group; CG, Control group; CI, Confidence Interval; *Md*, Mean difference; YBT, Y-balance test; 6MWT, six-minute walk test; cm, centimeter; kg, Kilogram.

Our 12-week Tai Chi training program demonstrated no significant differences in the mean upper-body strength scores of the either of the Tai Chi groups or the control group. During Tai Chi practice, the lower limbs primarily engage in anti-resistance exercises, while the upper limbs play a supportive role in coordination and balance. Previous studies by Lin et al. (2015) and Qi et al. (2020) did not report that Tai Chi improved upper-limb strength [42, 43],

**Table 4. Mean differences, *p*-value, and 95% confidence interval for differences for pairwise comparison after 12 weeks intervention among the three groups.**

| Parameters | BWTCG-CG Md (95% CI) | *p*-value | THTCG-CG Md (95% CI) | *p*-value | THTCG-BWTCG Md (95% CI) | *p*-value |
|---|---|---|---|---|---|---|
| Flexibility (cm) | 3.78(1.73 to 5.83) | < .001 | 4.67(2.62 to 6.72) | < .001 | 0.89(-1.16 to 2.94) | 0.391 |
| Upper limb strength (kg) | -3.22(-6.38 to -0.07) | 0.050 | -3.22(-6.38 to -0.07) | 0.050 | 0.01(-3.15 to 3.15) | 1.000 |
| Lower limb strength (second) | 26.82(13.07 to 40.56) | < .001 | 40.70(26.96 to 54.45) | < .001 | 13.89(0.14 to 27.64) | 0.048 |
| Single-leg stand-Stable ground (0–10 score) | -0.22(-0.85 to 0.41) | 0.482 | -0.70(-1.33 to -0.08) | 0.028 | -0.48(-1.11 to 0.15) | 0.130 |
| Single-leg stand-Soft cushion (0–10 score) | -1.04(-1.76 to -0.32) | 0.005 | -1.74(-2.46 to -1.02) | < .001 | -0.70(-1.43 to 0.02) | 0.056 |
| Tandem stand-Stable ground (0–10 score) | -0.48(-0.89 to -0.07) | 0.022 | -0.67(-1.08 to -0.26) | 0.002 | -0.19(-0.60 to 0.23) | 0.372 |
| Tandem stand-Soft cushion (0–10 score) | -0.67(-1.16 to -0.18) | 0.008 | -0.96(-1.45 to -0.47) | < .001 | -0.30(-0.79 to 0.19) | 0.231 |
| YBT(Right-leg)-Anterior (cm) | 5.15(2.02 to 8.28) | 0.002 | 7.70(4.57 to 10.83) | < .001 | 2.56(-0.57 to 5.69) | 0.108 |
| YBT(Right-leg)-Posteromedial (cm) | 6.56(2.82 to 10.29) | 0.001 | 9.52(5.79 to 13.25) | < .001 | 2.96(-0.77 to 6.70) | 0.118 |
| YBT(Right-leg)-Posterolateral (cm) | 8.44(4.37 to 12.52) | < .001 | 8.56(4.48 to 12.63) | < .001 | 0.11(-3.96 to 4.18) | 0.957 |
| YBT(Left-leg)-Anterior (cm) | 4.78(1.87 to 7.69) | 0.002 | 6.70(3.79 to 9.61) | < .001 | 1.93(-0.98 to 4.84) | 0.191 |
| YBT(Left-leg)-Posteromedial (cm) | 3.00(-0.65 to 0.65) | 0.106 | 7.04(3.38 to 10.69) | < .001 | 4.04(0.38 to 7.69) | 0.031 |
| YBT(Left-leg)-Posterolateral (cm) | 7.96(4.78 to 11.15) | < .001 | 10.33(7.15 to 13.52) | < .001 | 2.37(-0.81 to 5.55) | 0.142 |
| 6MWT (meter) | 44.04(11.33 to 76.74) | 0.009 | 59.41(26.70 to 92.11) | 0.001 | 15.37(-17.34 to 48.08) | 0.352 |

Note: Mixed design ANOVA test; Adjustment for multiple comparisons: Least Significant Difference.

BWTCG, Bafa Wubu Tai Chi group; THTCG, Traditional He-style Tai Chi group; CG, Control group; CI, Confidence Interval; *Md*, Mean difference; YBT, Y-balance test; 6MWT, six-minute walk test; cm, centimeter; kg, Kilogram.

and our research results also did not show any improvement. Given that Tai Chi emphasises relaxed and fluid movements without resistance, it is therefore possible that this Tai Chi movement characteristic factor might not have caused any changes to the participants' upper-limb strength.

The mean differences in lower-limb strength before and after the 12-week intervention were −27.56 seconds and −36.85 seconds for the BW-TC and TH-TC groups, respectively. A study by Lan et al. (2000) yielded similar results and demonstrated that healthy middle-aged individuals over 50 years of age who performed 60 minutes of Tai Chi every morning experienced an 9.6–18.8% increase in muscular endurance [44]. Our analysis found that the TH-TC group fared significantly better in improving the strength of their lower limbs than the BW-TC group; this finding is similar to the findings of Penn et al. (2019), who concluded that individualised Tai Chi exercises for the core and lower extremities improved lower-limb strength [9]. Tai Chi's continuous slow movements, knee flexion and weight-shifting require repeated resistance exercise that overcomes practitioners' own gravity and effectively enhances leg-muscle strength [44]. The TH-TC group fluctuation was higher than that of the BW-TC group, which may explain the superior result observed in the TH-TC group when compared to BW-TC group.

The measured walking distance for the 6MWT test before and after the BW-TC group and the TH-TCG group increased by 22.93 meters and 63.22 meters, respectively. The Tai Chi practice helped to improve the participants' muscle strength and endurance by requiring they keep their upper body upright, breathe sufficiently and convert their lower limbs' pace to supply oxygen and increase muscle strength; and the Tai Chi instructor followed these Tai Chi essentials and guided exercises for the two different Tai Chi groups, which played a key role in improving muscle strength and endurance. This is supported by a study by Caminiti et al. (2011), which reported a 36% increase in 6MWT walking distances following a Tai Chi training programme [45]. After 12 weeks of Tai Chi training, the BW-TC group and the TH-TC group increased their walking distance before and after the 6MWT test by 5.53% to 16.22%, respectively. While Caminiti et al. (2011) also observed Tai Chi improves the aerobic capacity of middle-aged and elderly practitioners, our study did not report any significant differences in the effect of practicing two types of Tai Chi on participants' aerobic endurance. Even though our Tai Chi training program included 30 minutes of continuous practice with an exercise intensity yielding a 50–70% maximum heart rate, our participants may not have fully benefited from the program.

Our study confirmed previous findings that Tai Chi practice can improve balance control [9, 20, 46–50]. The BW-TC and TH-TC groups both showed significant improvements in their static- and dynamic-balance scores; these results support other studies that have demonstrated the effectiveness of Tai Chi in improving balance [51]. The TH-TC group showed a greater increase in lower limb strength and better performance in the posteromedial side of their left legs, compared to the BW-TC group; while this could be due to differences in the Tai Chi training programmes, more research is needed to fully understand the physiological mechanisms behind these findings.

Tai Chi involves slow, repetitive movements that can increase lower-body muscle strength and flexibility, improve postural control and enhance balance [52]. According to Demnitz et al. (2018), Tai Chi may also contribute to cognitive improvement, because complex motor learning and practice are known to improve cognition [52]; as such, the complex movements of the BW-TC and TH-TC forms could therefore possibly improve the cognitive function of participating college students. Even though we did not evaluate the participants' cognitive ability in the present study, a study by Xiao et al. (2020) detected a correlation between Tai Chi practice and cognitive improvement [13].

While the present study validated the benefits of Tai Chi training, there are some limitations to consider. We did not evaluate the impact of Tai Chi on mental-health factors such as anxiety and depression. Furthermore, this study only included overweight male students, so it remains unclear whether the study findings can be applied to individuals with a healthy weight, those who are obese or females. Future research should therefore investigate the effects of these BW-TC and TH-TC programmes on practitioners' mental health and include a more diverse population and longer training periods.

## Conclusion

The current study demonstrated that both types of Tai Chi can effectively improve the physical fitness of overweight students, and neither was found to be superior to the other. Notably, however, Tai Chi did not improve the participants' upper-limb strength, and the TH-TC group more effectively improved their lower-limb strength. Consequently, these two styles of Tai Chi can serve as alternative exercise modalities to improve the physical fitness of overweight university students.

## Supporting information

**S1 File. Ethics.**
(PDF)

**S2 File. Research protocol.**
(PDF)

**S3 File. Reporting checklist for randomised trial.**
(DOCX)

**S4 File. Dataset.**
(XLSX)

**S5 File. Funding.**
(PDF)

## Acknowledgments

We wish to acknowledge all participants and staff who participated in this research activity and the support from Jiaozuo Normal University. We also wish to acknowledge the technical experts from the national physical health testing centre of Jiaozuo Tai Chi Sports Center for providing testing equipment services.

## Author Contributions

**Conceptualization:** Yantao Niu.

**Data curation:** Yantao Niu, Rojapon Buranarugsa.

**Formal analysis:** Yantao Niu, Rojapon Buranarugsa, Piyathida Kuhirunyaratn.

**Funding acquisition:** Yantao Niu.

**Investigation:** Yantao Niu.

**Methodology:** Yantao Niu, Rojapon Buranarugsa, Piyathida Kuhirunyaratn.

**Project administration:** Yantao Niu, Rojapon Buranarugsa.

**Resources:** Yantao Niu.

**Software:** Yantao Niu, Rojapon Buranarugsa, Piyathida Kuhirunyaratn.

**Supervision:** Yantao Niu, Rojapon Buranarugsa, Piyathida Kuhirunyaratn.

**Validation:** Rojapon Buranarugsa, Piyathida Kuhirunyaratn.

**Visualization:** Yantao Niu, Rojapon Buranarugsa, Piyathida Kuhirunyaratn.

**Writing – original draft:** Yantao Niu.

**Writing – review & editing:** Yantao Niu, Rojapon Buranarugsa, Piyathida Kuhirunyaratn.

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
