## [Decision Letter · Decision Letter 0]

19 Sep 2023

PONE-D-23-13273Effects of Bafa Wubu and He-Style Tai Chi exercise training on physical fitness of overweight university students: A randomized controlled trialPLOS ONE

Dear Dr. Buranarugsa,

Thank you for submitting your manuscript to PLOS ONE. After careful consideration, we feel that it has merit but does not fully meet PLOS ONE’s publication criteria as it currently stands. Therefore, we invite you to submit a revised version of the manuscript that addresses the points raised during the review process.

We look forward to receiving your revised manuscript.

Kind regards,

Hanna Landenmark

Staff Editor

PLOS ONE

Journal Requirements:

"We wish to acknowledge all participants and staff who participated in this research activity and the support from Jiaozuo Normal University. We also wish to acknowledge the technical experts from the national physical health testing centre of Jiaozuo Tai Chi Sports Center for providing testing equipment services. This work was supported by the 2022 Key University Research Project of the Department of Education of Henan Province, China grant number 22B890001."

**Additional Editor Comments:**

Please see the comments from two reviewers below. Please pay careful attention to the comments around the statistical analysis, and ensure that the conclusions accurately reflect the data.

Reviewers' comments:

Reviewer's Responses to Questions

**Comments to the Author**

1. Is the manuscript technically sound, and do the data support the conclusions?

Reviewer #1: Partly

Reviewer #2: Yes

2. Has the statistical analysis been performed appropriately and rigorously? 

Reviewer #1: No

Reviewer #2: Yes

3. Have the authors made all data underlying the findings in their manuscript fully available?

Reviewer #1: Yes

Reviewer #2: Yes

4. Is the manuscript presented in an intelligible fashion and written in standard English?

Reviewer #1: Yes

Reviewer #2: Yes

5. Review Comments to the Author

Reviewer #1: A three-arm randomized controlled trial was conducted which aimed to compare the effects of 12 weeks of Bafa Wubu Tai Chi (BW-TC) to traditional He-style Tah Chi (TH-TC) to control on physical fitness. The conclusions are unclear.

Major revisions:

1- Line 193-198: The ANOVA should be used to test the interaction effect of group by time. If the interaction effect is significant, provide an interpretation of the results, but do not test main effects because the tests for main effects are uninteresting in light of significant interactions. If interaction effects are non-significant, drop the interaction effects from the model and test the main effects. Determining which results to present when testing interactions is often a multi-step process. Specify the statistical method used for step-down testing when a significant interaction effect was observed.

2- Line 197-198: If the interaction test is conducted, the dependent t-tests are not necessary.

3- More clearly interpret the results associated with significant interaction effects.

Minor revisions:

1- Abstract: Specify the type of statistical summary values represented by -3.11/-4.52 cm, -27.56/-36.85 s, and -22.93/-63.22 m.

2- The statistical term for average is mean.

3- Lines 95-98: Provide complete details for estimating the sample size. The power calculation should include: (1) the estimated outcomes in each group; (2) the α (type I) error level; (3) the statistical power (or the β (type II) error level); (4) the target sample size and (5) for continuous outcomes, the standard deviation of the measurements. (6) The statistical testing method used.

4- Line 115: If block randomization was used, indicate the block size.

5- Line 190: Typographical error: presented.

6- Line 191: Specify the descriptive statistics. Indicate if the variables were checked for a normal distribution. Normally distributed variables are typically summarized using means and standard deviations while non-normally distributed variables are summarized with medians, first, and third quartiles.

7- Line 192: Replace “portray” with “summarize.”

8- Lines 207-210: To improve clarity, rearrange the presentation of results for time-by-group interaction effects. State those that were significant first.

9- P-values never equal zero; express small p-values as < 0.001.

10- Insert the tables where they should appear within the text.

Reviewer #2: Thanks for inviting me to review this wonderful work conducted by Rajapon and the rest of the team. This RCT examined the effects of two types of tai chi exercise on improving different physical fitness parameters in overweight university male students in mainland China during the COVID-19 pandemic. The authors identified a very interesting research gap in the current literature that there is a lack of head-to-head comparison between different types of tai chi on improving physical health. This trial was conducted of standard, and the conclusion was well supported by the results.

I have a few minor suggestions for the authors, listed as following:

1. Since the study participants are all male, I suggest the authors to indicate this in the study title.

2. In the abstract line 41 and 42, the improvements in sitting and reaching, wall squat, and 6-minute walk should be positive numbers? The minus symbol could be a mistake? Please check.

3. The study did not indicate which outcome is the primary outcome of the study, which is one of my major concerns, which is related to my next concern – sample size calculation.

4. In line 95 to 98, the authors briefly mentioned the method of sample size calculation, but the information about the effect size of the intervention(s) on the primary outcome, α, and power were missing. Were drop-out rates considered during the sample size calculation? Since this RCT was completed anyway, I suggest the authors to perform a power analysis to retrospectively investigate the power under the current sample size.

5. In the results section, were there any improvements in body weight and BMI? As the study participants were all overweight, this will be of the readers' interest to know.

Despite my concerns regarding the sample size calculation and primary outcome identification, I believe that this study offers valuable insights and innovative ideas to the field. I would suggest making minor revisions before publishing this work.

6. PLOS authors have the option to publish the peer review history of their article (what does this mean?). If published, this will include your full peer review and any attached files.

Reviewer #1: No

Reviewer #2: **Yes: **Danny J. Yu

---

## [Author Response · Author response to Decision Letter 0]

31 Oct 2023

Thank you to the reviewers and editor for the really helpful and constructive comments and providing us with the opportunity to make revisions to the manuscript to be considered for publication. We hope that by addressing the significant amount of feedback and comments from the reviewers, that you feel we have strengthened the manuscript. 

On the revised manuscript, Editor’s revisions are highlighted in red, Reviewer #1’s is highlighted in yellow, Reviewer 2’s is highlighted in blue. Responses to all the comments are in Respond to Reviewers document.

---

## [Decision Letter · Decision Letter 1]

21 Nov 2023

PONE-D-23-13273R1Effects of Bafa Wubu and He-Style Tai Chi exercise training on physical fitness of overweight university students: A randomized controlled trialPLOS ONE

Dear Dr. Buranarugsa,

Thank you for submitting your manuscript to PLOS ONE. After careful consideration, we feel that it has merit but does not fully meet PLOS ONE’s publication criteria as it currently stands. Therefore, we invite you to submit a revised version of the manuscript that addresses the points raised during the review process.

We look forward to receiving your revised manuscript.

Kind regards,

Miquel Vall-llosera Camps

Senior Staff Editor

PLOS ONE

Additional Editor Comments:

Please address the data analysis concerns as indicated by Reviewer#1 and modify your results and discussion accordingly.

Reviewers' comments:

Reviewer's Responses to Questions

**Comments to the Author**

1. If the authors have adequately addressed your comments raised in a previous round of review and you feel that this manuscript is now acceptable for publication, you may indicate that here to bypass the “Comments to the Author” section, enter your conflict of interest statement in the “Confidential to Editor” section, and submit your "Accept" recommendation.

Reviewer #1: (No Response)

2. Is the manuscript technically sound, and do the data support the conclusions?

Reviewer #1: Yes

3. Has the statistical analysis been performed appropriately and rigorously? 

Reviewer #1: No

4. Have the authors made all data underlying the findings in their manuscript fully available?

Reviewer #1: Yes

5. Is the manuscript presented in an intelligible fashion and written in standard English?

Reviewer #1: Yes

6. Review Comments to the Author

Reviewer #1: Major revisions:

The authors failed to make the distinction between interaction effects and main effects. Interaction effects include more than one variable. For instance time-by-group is an interaction effect. But the time effect alone is not an interaction effect; it is a main effect. I recommend that a qualified statistician be consulted to conduct these analyses and appropriately distinguish been interaction and main effects. Revise the wording in lines 203-211 (tracked changes version) to indicate that the standard practice for testing interactions has been conducted.

Standard Practice for testing interaction effects: If the interaction effect is significant, provide an interpretation of the results, but do not test main effects because the tests for main effects are uninteresting in light of significant interactions. If interaction effects are non-significant, drop the interaction effects from the model and test the main effects. Determining which results to present when testing interactions is often a multi-step process.

7. PLOS authors have the option to publish the peer review history of their article (what does this mean?). If published, this will include your full peer review and any attached files.

Reviewer #1: No

---

## [Author Response · Author response to Decision Letter 1]

3 Dec 2023

Thank you to the reviewers and editor for the really helpful and constructive comments and providing us with the opportunity to make revisions to the manuscript to be considered for publication. We hope that by addressing the significant amount of feedback and comments from the reviewers, that you feel we have strengthened the manuscript. We have now edited the data analysis presentation again based on the reviewers' comments.

---

## [Decision Letter · Decision Letter 2]

29 Dec 2023

Effects of Bafa Wubu and He-Style Tai Chi exercise training on physical fitness of overweight university students: A randomized controlled trial

PONE-D-23-13273R2

Dear Dr. Buranarugsa,

We’re pleased to inform you that your manuscript has been judged scientifically suitable for publication and will be formally accepted for publication once it meets all outstanding technical requirements.

Kind regards,

Miquel Vall-llosera Camps

Staff Editor

PLOS ONE

Reviewers' comments:

Reviewer's Responses to Questions

**Comments to the Author**

1. If the authors have adequately addressed your comments raised in a previous round of review and you feel that this manuscript is now acceptable for publication, you may indicate that here to bypass the “Comments to the Author” section, enter your conflict of interest statement in the “Confidential to Editor” section, and submit your "Accept" recommendation.

Reviewer #1: All comments have been addressed

2. Is the manuscript technically sound, and do the data support the conclusions?

Reviewer #1: (No Response)

3. Has the statistical analysis been performed appropriately and rigorously? 

Reviewer #1: (No Response)

4. Have the authors made all data underlying the findings in their manuscript fully available?

Reviewer #1: (No Response)

5. Is the manuscript presented in an intelligible fashion and written in standard English?

Reviewer #1: (No Response)

6. Review Comments to the Author

Reviewer #1: All comments have been adequately addressed.

7. PLOS authors have the option to publish the peer review history of their article (what does this mean?). If published, this will include your full peer review and any attached files.

Reviewer #1: No

---

## [Editor Report · Acceptance letter]

9 Jan 2024

PONE-D-23-13273R2 

PLOS ONE

Dear Dr. Buranarugsa, 

I'm pleased to inform you that your manuscript has been deemed suitable for publication in PLOS ONE. Congratulations! Your manuscript is now being handed over to our production team.

Kind regards, 

on behalf of

Dr. Miquel Vall-llosera Camps 

Staff Editor

PLOS ONE